# MEEP: Is this Engaging? Prompting Large Language Models for Dialogue Evaluation in Multilingual Settings

**Amila Ferron** and **Amber Shore** and **Ekata Mitra** and **Ameeta Agrawal**

Department of Computer Science, Portland State University

{aferron,ashore,ekata,ameeta}@pdx.edu

## Abstract

As dialogue systems become more popular, evaluation of their response quality gains importance. Engagingness highly correlates with overall quality and creates a sense of connection that gives human participants a more fulfilling experience. Although qualities like coherence and fluency are readily measured with well-worn automatic metrics, evaluating engagingness often relies on human assessment, which is a costly and time-consuming process. Existing automatic engagingness metrics evaluate the response without the conversation history, are designed for one dataset, or have limited correlation with human annotations. Furthermore, they have been tested exclusively on English conversations. Given that dialogue systems are increasingly available in languages beyond English, multilingual evaluation capabilities are essential. We propose that large language models (LLMs) may be used for evaluation of engagingness in dialogue through prompting, and ask how prompt constructs and translated prompts compare in a multilingual setting. We provide a prompt-design taxonomy for engagingness and find that using selected prompt elements with LLMs, including our comprehensive definition of engagingness, outperforms state-of-the-art methods on evaluation of engagingness in dialogue across multiple languages.[1]

## 1 Introduction

Dialogue systems are becoming more popular, but their quality is usually evaluated in terms of metrics such as fluency, coherence, or sensibility. Recent advancements in large language model-based dialogue systems enable high levels of proficiency, thus shifting the emphasis from fluency evaluation to more nuanced aspects such as engagingness, which has emerged as an important quality of dialogue systems (Yu et al., 2016; Zhou et al., 2022;

---

[1]Code available at https://github.com/PortNLP/MEEP

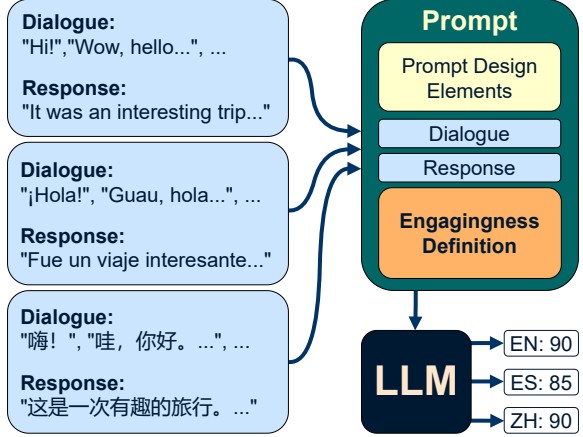

Figure 1: Overview of proposed methodology, MEEP, on datasets in English, Spanish, and simplified Chinese. Our prompt is distinguished by its definition of engagingness, drawn from previous research, linguistic analysis, and the study of inclusivity. Extensive experiments informed the use of additional prompt design elements. Scores shown here are in a range from 0 (not engaging) to 100 (very engaging).

Cohen et al., 2022). Development of models that produce more engaging responses can be supported by automatic metrics to complement human annotation.

Despite the ground-breaking work of existing metrics for engagingness, they evaluate the response without the conversation history (Xu et al., 2022), or are designed for a specific dataset (Liu et al., 2023), and are not highly correlated with human annotations (Ghazarian et al., 2020). Although multi-dimensional automatic metrics are desirable, they have limited success with complex qualities like engagement (Mehri and Eskenazi, 2020a; Deng et al., 2021; Zhang et al., 2022a). Enthusiasm for multi-dimensional evaluation is balanced by calls to develop metrics that measure specific dialogue qualities in order to complement existing metrics (Gehrmann et al., 2022; Mehri et al., 2022; Colombo et al., 2022).

Furthermore, dialogue systems are increasingly available in languages beyond English so it is important to be able to test systems in these languages (Rodríguez-Cantelar et al., 2023). With the rise of virtual assistants, the importance of engaging dialogue (Richardson et al., 2023) and multilingualism has increased significantly, as they need to actively assist users, facilitate knowledge exploration, and foster effective communication, making engagingness a vital parameter in evaluating dialogues. Our research therefore develops an automatic metric for engagingness that can evaluate dialogue responses in multiple languages.

We establish a comprehensive definition of engagingness in conversation and construct an extensive range of prompts designed with our dimensions of engagingness and prompt engineering techniques (see Figure 1). These prompts are employed across several LLMs, which show impressive capabilities in many tasks and have had limited exploration into their use for automatic evaluation of dialogue.

To our knowledge, this is the first work to extensively test prompt design for the dedicated evaluation of engagingness in dialogue in a multilingual setting. We test the prompts first on English-only datasets, and select the best-performing set of prompting methods to test against four strong baseline metrics on Spanish-language and simplified Chinese-language datasets. We also test translated versions of these prompts. We find that our proposed framework – MEEP: Metric for Engagingness Evaluation using Prompting – can be effective for evaluation of enganginess in dialogue in multiple languages with the broader implication that LLMs can be used for evaluation of complex qualities of dialogue in multiple languages through prompting alone.

Our contributions are as follows:

- A thorough definition of engagingness using five dialogue engagement dimensions informed by previous engagingness research and a linguistics-based approach. It frames the goals of engagingness in conversation in terms of human-aligned dimensions independent of existing datasets and provides a more nuanced target for the development of engaging dialogue systems.

- A novel method for measuring engagingness in dialogue using a set of prompt engineering approaches that improve LLM performance.

- An extensive evaluation across ten datasets and three languages (English, Spanish, and Chinese).

## 2 Related Work

**Automatic Dialogue Metrics**    From the development of PARADISE, the earliest automatic metric to evaluate dialogue (Walker et al., 1997), reference-based automatic metrics such as BLEU (Papineni et al., 2002), ROUGE (Lin, 2004), and BERTSCORE (Zhang et al., 2020) have been used extensively for evaluation of dialogue but have weak correlation with human judgments (Liu et al., 2016; Novikova et al., 2017), and do not suit the multiplicity of valid responses possible in any conversation (Zhao et al., 2017; Mehri et al., 2022).

**Automatic Metrics for Engagingness in Dialogue** Ghazarian et al. (2020) use engagingness as a measure of dialogue system performance in the first attempt to learn engagingness scores at the utterance level from annotated data. This BERT-based metric achieves modest correlations with human annotated scores, and leaves room for future improvement. A recent effort at creating a more robust automatic metric for engagingness is ENDEX (Xu et al., 2022). This method uses several features of a Reddit-based dataset to learn engagingness scores and evaluate the responses in a turn-level dataset without considering the dialogue context. Some multi-dimensional dialogue metrics include engagingness as one of the dimensions measured (Mehri and Eskenazi, 2020a; Zhang et al., 2021), however, the engagingness scores for these metrics are not highly correlated with human evaluation. Other multidimensional dialogue metrics use results from human-directed questions about interestingness to evaluate engagingness for an imprecise approximation (Deng et al., 2021; Zhong et al., 2022; Liu et al., 2023).

**Multilingual Automatic Dialogue Metrics** Multilingual dialogue models are commonly evaluated using similarity measures such as SacreBLEU, BERTSCORE, BLEU, and perplexity (Agarwal et al., 2021; Zhang et al., 2022b). Development of non-similarity-based automatic metrics for a multilingual context remains an open question for research. The recent DSTC11 shared task proposal for Task 4 acknowledges this lack as motivation for the task (Rodríguez-Cantelar et al., 2023).

**Large Language Models for NLG Evaluation**

Gao et al. (2023) find that for text summarization evaluation, ChatGPT provides scores in the style of human evaluation, and that adding dimension definitions leads to slight improvements in correlation with human annotations. GEMBA (Kocmi and Federmann, 2023) achieves state-of-the-art performance for machine translation evaluation using GPT-3.5 and larger. Most attempts to study the effectiveness of prompting in the context of automatic evaluation use modified versions of annotator instructions as prompts (Kocmi and Federmann, 2023; Wang et al., 2023; Luo et al., 2023; Fu et al., 2023; Gao et al., 2023). Prompt techniques improve through the use of clarifying examples (Yuan et al., 2021) or assigning LLMs roles (Shen et al., 2023). Svikhnushina and Pu (2023) use new dialogues generated by an LLM interacting with a chatbot that are then scored with the same LLM.

**Large Language Models for Evaluation of Dialogue** G-EVAL (Liu et al., 2023), evaluates summarization and dialogue response generation, using prompts that include definitions of the task and quality to be evaluated, Chain of Thought reasoning, and a scoring function. GPTSCORE (Fu et al., 2023) provides an adaptable metric using zero-shot instruction with a prompt template that includes the task specification and a definition of the aspect to be evaluated. Each has only been tested on one (English) dataset.

## 3 Engagingness in Dialogue

Engagingness is central to our conception of conversation, as demonstrated by the prevalence of the word "conversation" as a collocate[2] of the verb "engage" (COCA). What then is engagingness? In prior research, several metrics conflate engagingness with interestingness (Deng et al., 2021; Zhong et al., 2022; Liu et al., 2023). Others provide no definition and the training of these metrics rely instead on the implicit judgements from the training set annotators (Logacheva et al., 2018). We posit that there is no standard definition of engagingness in NLP because of the subjectivity of human judgment. Humans, like models, cannot be relied upon for self reporting (Gao et al., 2021; Rosenman et al., 2011). We put forward a definition of engagingness that is useful for our goal: to create an evaluation tool that will improve dialogue model performance.

---

[2]A collocate is a word that appears in conjunction (or in the same context) as the other word with greater than random frequency.

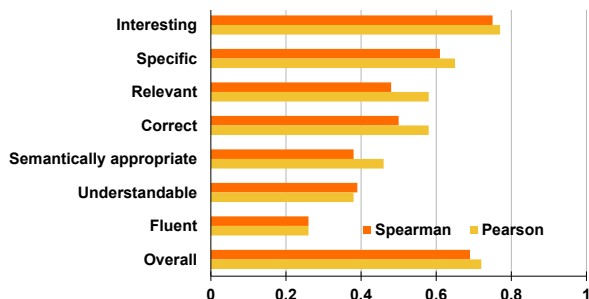

Figure 2: Correlation of several dialogue dimensions with engagingness as annotated in the FED dataset.

To ground our definition, we start with the dictionary. Dictionary definitions identify three consistent aspects of the quality of being engaging: *attention*, *interest*, and *participation* (Merriam-Webster, 2023; Wiktionary, 2023; Dictionary, 2023). A reply is engaging when it gets the user's attention, interests the user, and incites participation from the user. These three aspects form the core of what we expect an engagingness metric to measure.

We propose the following five subdimensions of engagingness:

- **Response Diversity:** Engaging responses are diverse, not repetitive and generic. When a model produces a standard or generic response, it can constrain the responses the user can reasonably produce (Stasaski and Hearst, 2023), creating predictable dialogue which can limit engagingness.

- **Interactional Quality:** Engaging responses encourage a response from the user. A definitionally necessary component of engagement is participation, so signs of elicited participation in the user, such as 1) the presence of any response, and 2) the presence of a high-quality response, demonstrate this aspect.

- **Interestingness:** We pose interestingness as a necessary component of engagingness. Research has commonly used interestingness as a proxy for engagingness (Deng et al., 2021; Zhong et al., 2022; DSTC11 2023; Liu et al., 2023), and we see high correlation between scores for engagingness and interestingness in the FED dataset (Mehri and Eskenazi, 2020a) (see Figure 4). However, we do not believe that there is a one-to-one correspondence between interestingness and engagingness. Interestingness captures factual appeal (Mehri and Eskenazi, 2020b), while definitions of engagingness emphasize active participation. Both

| Prompt Code | Description |
|---|---|
| Naive | simple baseline prompt; directly asking for a score |
| Naive+R | Naive prompt with role assignment |
| HD | prompt derived from instructions given to human annotators of engagingness |
| HD+R | the human-directed prompt with role assignment |
| MEEP | an intro like the HD prompt, and short phrases for each of our six subdimensions of engagingness |
| MEEP+SA | the MEEP prompt with the "such as" phrase |
| MEEP+R | the MEEP prompt with role assignment |
| MEEP+SA+R | the MEEP prompt with the "such as" phrase and role assignment |
| MEEP+SA-DIAL | the MEEP+SA prompt edited to be used at the dialogue-level |
| MEEP+SA+R-DIAL | the MEEP+SA+R prompt edited to be used at the dialogue-level |

Table 1: Overview of prompt styles. Some prompts are also translated into other languages.

contribute to meaningful conversations from different perspectives.

- **Contextual Specificity:** Engaging responses are specific to the conversational context. This is another aspect of avoiding generic user responses. See et al. (2019) show that controlling for specificity improves the engagingness of their dialogue model.

- **Othering:** Engaging responses create a sense of belonging, not one of othering (Powell and Menendian, 2022; Alexander and Andersson, 2022). Even an interesting conversation can quickly turn dull when we feel misunderstood, unheard, and looked down on. Conversations where rapport is established, on the other hand, are often enjoyable even in the absence of content. This aspect of engagingness is particularly underexplored in the research.

## 4 MEEP: Metric for Engagingness Evaluation using Prompting

We design several categories of prompts. A straightforward naive prompt acts as the basis against which we compare our more developed prompt sets. We adapt dialogue annotator directions into prompts for the human-directed prompt set and our own proposed sub-dimensions of engagingness form the third prompt set. For each prompt set, we create a version casting the LLM in a role. Table 1 provides an overview of the prompt types, and the full text of each prompt is available in Appendix A. We experiment with prompts in English as well as Spanish and Chinese.

**Naive Prompt (Naive)** We use the **Naive** prompt as a baseline comparator from which to develop more performant prompts. Kocmi and Federmann (2023) report best performance from the least constrained prompting method, implying that this naive prompt approach is valid as its own path of inquiry, and a similar prompt is used in Wang et al. (2023).

**Human-directed Prompts (HD)** Since LLMs are developed to approximate natural language and natural linguistic interaction, we theorize that an evaluation prompt styled after the same instructions that human annotators were given should lead to increased performance over the Naive prompt.

**Prompts With Our Proposed Dimensions of Engagingness (MEEP)** We incorporate each of our five subdimensions of engagingness into a word or phrase in order to form the definition of engagingness within the prompt. This is demonstrated in Figure 3.

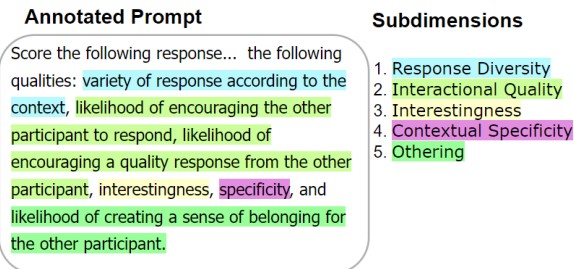

Figure 3: The MEEP prompt annotated to show the five engagingness subdimensions.

**"Such As" and Role Assignment (SA, R)** Yuan et al. (2021) find that even when controlling for the presence of example responses, the phrase "such as" increases performance in the domain of machine translation evaluation. When added to our prompts, it is accompanied by an example response. Role assignment is theorized to increase the quality of the responses generated by dialogue models (Svikhnushina and Pu, 2023), and we explore its utility in dialogue response evaluation. We incorporate role assignment using methods specific to the model used. When possible, we make the assignment directly in the API call, using the "system" option. If this method is not supported by

the model, the role assignment is prefixed to the prompt text.

**Translation (†)** We translate a selection of our prompts from English into Spanish and Chinese, to match the language of the dialogues evaluated. Spanish text was manually translated, and Chinese translations are with GPT-4 and checked by a human translator for quality.

# 5 Experiments

We first test our prompt combinations using two LLMs on the English-language FED dataset. Five examples from these tests are randomly selected, on which we perform a qualitative analysis. We then select the best performing prompts to test over six multilingual turn-level datasets, two in each of English, Spanish, and Chinese. For these tests, we evaluate the prompts in English as well as the versions translated into the language of the dataset. The scores reported by each LLM/prompt style pair over six datasets are correlated with the human-annotated scores to evaluate the proposed methods and four strong baseline methods. A final set of experiments evaluates the performance of the highest-performing prompts on dialogue-level datasets in English and Chinese and correlates with human-annotated scores. These are compared with the strongest baseline method from turn-level testing.

## 5.1 Large Language Models

For our base model, we select `text-davinci-003` (GPT-3.5) (Ouyang et al., 2022), `gpt-3.5-turbo-0301` (ChatGPT), and `gpt-3.5-turbo-0613` (ChatGPT0613) from the GPT-series of LLMs[3] (Radford et al., 2022). We set the temperature to 0, `top_p` to 1, and `n` to 1. Presence penalty and frequency penalty are set to 0, `logit bias` to null, and we request the `best of 1`. We also use LLaMA2 (Touvron et al., 2023) in the 7B size with the `chat_completion` function with `max_seq_len` set to 1024, `max_gen_len` set to None, temperature set to 0, `max_batch_size` set to 1, and `top_p` set to 1. We do not set a max length for generation because the responses are long and the location of scores within the returned text is unpredictable.

## 5.2 Benchmark Datasets

We use four existing datasets, all from the 11th Dialogue System Technology Challenge, Track 4[4] (DSTC11 2023). Challenge organizers machine translated two English turn-level datasets into Spanish and Chinese, and two Chinese dialogue-level datasets into English to produce a total of ten datasets spanning three languages. Annotations from the original datasets are used to test the datasets in the additional languages.

**FED** The Fine-grained Evaluation of Dialogue (FED) dataset (Mehri and Eskenazi, 2020a) provides annotations of English-language dialogues first collected by Adiwardana et al. (2020) by way of multi-dimensional evaluations per dialogue turn, including engagingness as one of the evaluated dimensions.

**SEE** The Persona-See (SEE) dataset (See et al., 2019) expands the PersonaChat (Zhang et al., 2018) task to provide human annotations at the turn-level, evaluating multiple dimensions of conversation per dialogue. They consider enjoyment to be an equivalent measure of engagement, relying on users' active involvement and positive perception. We regard it as a reliable proxy for engagingness. Originally containing over 3,000 annotated dialogues, this is much larger than the other datasets, seen in Table 2. For efficiency and consistency of scale, we randomly select a subset of 300 dialogues for comparative analysis. These same 300 dialogues are used in each of English, Spanish, and Chinese.

**KDCONV** The Knowledge-driven Conversation (KDCONV) dataset consists of Chinese-language dialogues between two humans who are given knowledge graphs from which to draw their responses (Zhou et al., 2020).

**LCCC** The large-scale cleaned Chinese conversation dataset (LCCC) contains conversations from Chinese social media posts of users heuristically identified to be human (Wang et al., 2020). These dialogues are combined with those from publicly available databases and cleaned.

Dialogues for KDCONV and LCCC were manually annotated at the dialogue-level for engagingness and machine translated into English for the DSTC11 Track 4 Challenge.

**Translation Quality Analysis** We briefly look at dataset translation quality to assess the validity of

---

[3]We note that ChatGPT is available at the time of writing as `gpt-3.5-turbo` in the free version, and `gpt-4` in the paid version. We take the liberty of using ChatGPT as the shorthand for `gpt-3.5-turbo-0301`, the free version at the time of testing.

[4]https://chateval.org/dstc11

| Dataset | # samples | Language | Annotation level | COMET-20 | COMET-21 | CosSim1 | CosSim2 |
|---------|-----------|----------|------------------|----------|----------|---------|---------|
| FED-EN | 375 | English | turn | – | – | – | – |
| FED-ES | 375 | Spanish | turn | 0.458 | 0.117 | 0.846 | 0.914 |
| FED-ZH | 375 | Chinese | turn | 0.323 | 0.084 | 0.789 | 0.883 |
| SEE-EN | 300 | English | turn | – | – | – | – |
| SEE-ES | 300 | Spanish | turn | **0.510** | **0.128** | **0.873** | **0.921** |
| SEE-ZH | 300 | Chinese | turn | 0.376 | 0.102 | 0.800 | 0.875 |
| KDCONV-EN | 392 | English | dialogue | 0.242 | 0.053 | 0.788 | 0.857 |
| KDCONV-ZH | 392 | Chinese | dialogue | – | – | – | – |
| LCCC-EN | 22 | English | dialogue | -0.132 | -0.40 | 0.690 | 0.776 |
| LCCC-ZH | 22 | Chinese | dialogue | – | – | – | – |

Table 2: Dataset specifications, with average machine translation quality scores of datasets that were translated. The highest scores for each measure are in **bold**. Second highest scores are underlined.

the annotations as an accurate measure of engagingness for translated datasets. This informs analysis of our test results in Section 6.2. Translations into Spanish and Chinese were obtained by the DSTC11 organizers with the MS Azure service[5] and Tencent Machine Translation service[6], respectively.

The datasets include four measures of translation quality for every utterance: two quality estimator metrics from COMET versions from 2020 (Rei et al., 2020) and 2021 (Rei et al., 2021), and two cosine similarity measures from embeddings generated from the original utterance and translation. Two methods of embedding generation were used. Details are available in Appendix B. We average the provided translation quality scores across all utterances for each translated dataset and list them in Table 2. The Spanish-language datasets score consistently higher than the Chinese-language datasets.

### 5.3 Baselines

The results of our proposed method are compared against four recent baselines.

**ENDEX** evaluates the engagingness of dialogues based on human reactions (Xu et al., 2022). This approach uses fine-tuned RoBERTa-large models on datasets evaluated at the turn level, measuring dimensions of engagement including attentional, behavioral, emotional, and task-based aspects of replies. Each response is given a binary score for engagingness. Due to the inherent constraints of ENDEX, we encountered difficulties in adapting it to languages other than English.

**UNIEVAL** employs T5-Large as the underlying model to assess any dialogue dimension, including those that are not seen during training (Zhong et al., 2022). It accomplishes this by formulating single evaluation dimensions as boolean question-answering (QA) tasks. In evaluating the engagingness of a response, UNIEVAL takes three parameters: the conversational context, an additional context (such as an intriguing fact), and the response itself. To facilitate testing with the FED dataset which does not include an additional fact, an empty string is used as the additional context. We use UNIEVAL in a multilingual context where it had not been previously evaluated.

**GPTSCORE** evaluates dialogue response generation at the turn-level across eight dimensions, including engagingness (Fu et al., 2023). Their prompt for evaluation of engagingness is included in Appendix A. Our approximation of their methodology is adapted from their code for text summarization evaluation. A prompt that includes task definition, identification of the quality to be evaluated (engagingness), and the dialogue, followed by "Answer: Yes.", is passed to the LLM with `temperature=0`, `max_tokens=0`, `logprobs=0`, `echo=True`, and `n=0`. The log probabilities returned for the token 'Yes' become the score for engagingness.

**G-EVAL** is an LLM-based evaluator comprising a prompt, Chain-of-Thought reasoning, and a scoring function (Liu et al., 2023). It achieves strong correlations with human evaluations using GPT-3.5 and GPT-4 models in NLG tasks, including the assessment of engagingness in dialogues. We use their GPT-3.5 model, which shows higher correlations, and modify the prompt for testing datasets without additional context, as seen in Appendix A.

---

[5] https://azure.microsoft.com/en-us/products/cognitive-services/translator/

[6] https://www.tencentcloud.com/products/tmt

| Prompt | GPT3.5 | | ChatGPT | | ChatGPT0613 | | LLaMA2-7B | | Average | |
|---|---|---|---|---|---|---|---|---|---|---|
| | S | P | S | P | S | P | S | P | S | P |
| Naive | 0.445 | **0.521** | 0.434 | 0.463 | 0.416 | 0.435 | 0.356 | 0.272 | 0.413 | 0.423 |
| Naive+R | 0.416 | 0.466 | 0.464 | 0.469 | 0.471 | 0.480 | 0.356 | **0.322** | 0.427 | 0.434 |
| HD | 0.509 | 0.511 | 0.462 | 0.472 | 0.511 | 0.535 | -0.007* | 0.046* | 0.369 | 0.391 |
| HD+R | 0.492 | 0.500 | 0.497 | 0.508 | 0.518 | 0.526 | 0.112 | 0.029* | 0.405 | 0.391 |
| MEEP | 0.475 | 0.508 | 0.548 | 0.553 | 0.511 | 0.504 | 0.333 | 0.111 | 0.467 | 0.419 |
| MEEP+R | 0.494 | 0.507 | 0.540 | 0.543 | 0.514 | 0.516 | 0.402 | -0.057* | 0.487 | 0.377 |
| MEEP+SA | **0.532** | 0.520 | 0.558 | **0.573** | 0.537 | **0.549** | **0.410** | 0.091* | **0.509** | 0.433 |
| MEEP+SA+R | 0.526 | 0.517 | **0.568** | 0.566 | **0.541** | 0.548 | 0.376 | 0.113 | 0.503 | **0.436** |

Table 3: Results for prompts: Naive, Human-Directed (HD), and MEEP, with ablation results for prompt elements *such as* (SA) and system role (R). Spearman (S) and Pearson (P) results are listed for each experiment. Results marked with * are not statistically significant. The highest result for each column is in **bold**, with the second highest result underlined. The FED-EN dataset was used for testing.

## 6 Results and Analysis

### 6.1 Effect of Prompt Styles

**Effect of MEEP** Average correlations of system output and human annotations for each prompt are shown in Table 3. Prompts including MEEP generally give higher correlation than experiments not including MEEP in the prompts. The method with lowest correlation is the human-directed prompt, with average correlations of 0.420 (S) and 0.429 (P). The MEEP-based prompts have average correlations of 0.492 (S) and 0.481 (P). This indicates that including our fine-grained subdimensions of engagingness in prompting for evaluation of engagingness is an important component of an effective evaluation method.

**Effect of 'such as'** As seen in Table 3, prompts with our engagingness dimensions (MEEP) and the use of 'such as' have the highest and second highest correlations, in 5 of 8 measures. Prompts including 'such as' have average correlations of 0.506 (S) and 0.492 (P). Their performance dominates both with and without defining the system role.

**Effect of system role** In Table 3, we see that the use of system role has mixed impact. Performance for ChatGPT and ChatGPT0613 improves with system role in most cases, while performance varies for LLaMA2-7B and generally falls for GPT-3.5. Defining the system role gives the most consistent gains in performance when used with ChatGPT on prompts that do not include MEEP. The inconsistent improvements with the use of system role may be because it offers a contextual shift that is not as closely aligned as other methods, like MEEP. This more approximate alignment at times moves the model to a less-aligned position than it would be if using a better aligned method alone. GPT-3.5

almost entirely performs worse with the prepended system role, indicating that the system role prompt is not beneficial when used with GPT-3.5 in this setting. This combined with mixed performance when used on LLaMA2-7B suggests that it may add a level of complexity to which the simpler models cannot rise.

**Comparing Models** Considering that ChatGPT is an interactive LLM specifically trained on conversational data, we are curious to see whether it brings additional gains to our task. In Table 3, we see that ChatGPT has an average of 0.023 (S) and 0.012 (P) higher correlations than GPT-3.5 across all English-language turn-level experiments. ChatGPT0613 has similar though slightly lower aggregate performance to that of ChatGPT. LLaMA has 0.151 (S) and 0.302 (P) lower average performance than GPT3.5, considering statistically significant results only. GPT-3.5 has better performance than ChatGPT and ChatGPT0613 for the simplest prompts, when used without system role assignment, otherwise ChatGPT is the model that is most aligned with human annotations.

### 6.2 Performance in a Multilingual Setting

#### 6.2.1 Turn-level Evaluation

Table 4 reports Spearman coefficients from experiments on multilingual turn-level datasets with prompts in English, Spanish, and Chinese (Pearson coefficients are available in Appendix C). Our highest-performing English prompts have an average of 0.064 higher correlation than the highest-performing baseline metric (G-EVAL or GPTSCORE) for that dataset. Average increases of the highest-correlated English prompt over the highest-correlated baseline are 0.060, 0.082, and 0.050 for the English, Spanish, and Chinese-language datasets, respectively.

| Model | Prompt | FED | | | SEE | | |
|---|---|---|---|---|---|---|---|
| | | EN | ES | ZH | EN | ES | ZH |
| ENDEX | – | 0.290 | – | – | 0.164 | – | – |
| UNIEVAL | – | 0.190 | 0.258 | 0.076* | 0.015* | 0.073* | 0.038* |
| GPTSCORE | GPTSCORE | 0.176 | 0.146 | 0.230 | 0.087* | 0.153 | 0.140 |
| G-EVAL | G-EVAL | 0.488 | 0.448 | 0.402 | 0.194 | 0.131 | 0.062* |
| GPT-3.5 | MEEP+SA | 0.532 | 0.481 | 0.451 | **0.236** | 0.223 | 0.189 |
| ChatGPT | MEEP+SA | 0.558 | 0.516 | 0.431 | 0.169 | 0.138 | 0.133 |
| ChatGPT | MEEP+SA+R | **0.568** | **0.542** | 0.435 | 0.160 | 0.150 | 0.140 |
| ChatGPT-0613 | MEEP+SA+R | 0.550 | 0.471 | 0.400 | 0.214 | 0.200 | 0.175 |
| GPT-3.5 | *MEEP+SA†* | – | 0.438 | **0.520** | – | 0.128 | 0.085* |
| ChatGPT | *MEEP+SA†* | – | 0.500 | 0.408 | – | 0.161 | 0.149 |
| ChatGPT | *MEEP+SA+R†* | – | 0.525 | 0.444 | – | 0.123 | 0.168 |
| ChatGPT-0613 | *MEEP+SA+R†* | – | **0.542** | 0.374 | – | **0.273** | **0.227** |

Table 4: Correlation results using Spearman coefficient for multilingual turn-level experiments. All results are statistically significant except those labeled with *. '†' denotes a translated version of the prompt into the language of the dataset. Highest results are in **bold**.

| Model | Prompt | KDCONV | | LCCC | |
|---|---|---|---|---|---|
| | | EN | ZH | EN | ZH |
| G-EVAL | G-EVAL-DIAL | **0.327*** | 0.189 | 0.149 | 0.248 |
| GPT-3.5 | MEEP+SA-DIAL | 0.286* | 0.223* | **0.282** | **0.287** |
| ChatGPT | MEEP+SA-DIAL | 0.178* | 0.392* | 0.073* | 0.191 |
| ChatGPT | MEEP+SA+R-DIAL | 0.185* | **0.428** | -0.004* | 0.193 |
| ChatGPT-0613 | MEEP+SA+R-DIAL | 0.186* | 0.362* | 0.032* | 0.117 |

Table 5: Correlation results using Spearman coefficient for multilingual dialogue-level experiments. Results that are not statistically significant are marked with *. Highest scores for each dataset are in **bold**.

Results indicate that correlation using our method is highest for English-language dialogues, followed by Spanish-language dialogues, with Chinese-language dialogues having the lowest correlations. We would expect to see comparable performance because they are all high-resource languages. The unexpected results may be related to translation quality of the datasets, which is lower for the Chinese datasets as seen in Table 2. A comparison of results for the Spanish and Chinese datasets in Figure 4 shows that translating a prompt into the language of the dataset does not consistently improve correlation, but our best scores for the Spanish and Chinese datasets are nevertheless seen with translated prompts. The ChatGPT0613 model provides several of these highest correlations, indicating that improved system role capabilities with this model may have included multilingual system role training.

### 6.2.2 Dialogue-level Evaluation

Results for dialogue-level evaluation are in Table 5. Achieving statistically significant results is less consistent on the dialogue-level than on the turn-level datasets, especially for the translated English-language versions. Despite this, we can see better performance with our prompts than with the baseline.

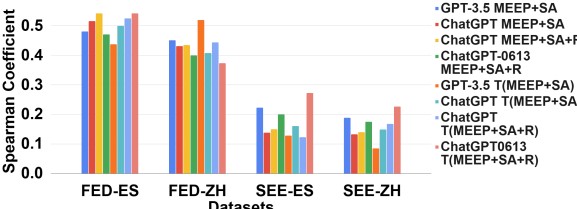

Figure 4: Correlation of each prompt on the Spanish and Chinese datasets.

**Qualitative Analysis** An example from qualitative analysis is provided in Table 6, with more examples in Appendix D. The randomly-selected examples show mixed consistency with general trends observed with testing. We note that MEEP prompts improve performance over aggregate results from HD and Naive prompts, although exceptions can be found. The effect of the system role is difficult to discern at this scale. ChatGPT performs better or similarly to GPT-3.5 in four out of five examples, holding with the pattern exhibited in the data.

We see a pattern of GPT-3.5 more often giving scores that are higher than the ground truth annotations. When we average the scores for the top-performing models, GPT-3.5 and ChatGPT, across all the model/prompt combinations used in Table 3, we find that GPT-3.5 produces scores appreciably higher than ChatGPT, with averages of 0.8138 and

| Dialogue: "Hi!" |
|---|
| "Hiii! How are you?" |
| "I am good" |
| ... |
| "You're welcome! How long until you're done with training models?" |
| "We keep on trying to improve them so I guess it'll be a while" |
| Response: "That's understandable. Good luck, I hope it goes smoothly!" |

Normalized average of human annotations: **0.6**

| Prompt | GPT-3.5 | $\delta$ | ChatGPT | $\delta$ |
|---|---|---|---|---|
| Naive | 0.9 | 0.3 | 0.85 | 0.25 |
| Naive + R | 1.0 | 0.4 | 0.85 | 0.25 |
| HD | 1.0 | 0.4 | 0.5 | 0.1 |
| HD+R | 1.0 | 0.4 | 0.5 | 0.1 |
| MEEP | 0.9 | 0.3 | 0.8 | 0.2 |
| MEEP+R | 0.9 | 0.3 | 0.8 | 0.2 |
| MEEP+SA | 0.8 | 0.2 | 0.8 | 0.2 |
| MEEP+SA+R | 0.9 | 0.3 | 0.8 | 0.2 |

Table 6: Comparison of results for one example dialogue from the FED dataset for qualitative analysis. The distance from the ground truth ($\delta$) for each score is listed to the right of the score.

0.7307, respectively. This indicates that GPT-3.5 has a positive bias in evaluation when compared to ChatGPT, which may reveal underlying limitations in the model's capability.

## 7 Discussion

Insights from our experiments showcase the power of our proposed LLM-based evaluator as follows:

- Our MEEP dimensions of engagingness improve alignment with human annotations and are an effective component of an LLM-based metric, improving over state-of-the-art metrics in evaluation of engagingness.

- Clear improvement is seen with the use of *such as* with clarifying examples in this context and we conclude that examples written in this form will improve dialogue evaluation performance with these LLMs.

- Defining the system role offers improvement when used with ChatGPT or ChatGPT0613 and prompts that are naive. It does not improve performance when used with LLaMA2-7B, or GPT-3.5.

- ChatGPT performs better than GPT-3.5 with more complex prompts. When prompts are in their simplest form, GPT-3.5 has higher correlation with human annotations. LLaMA gives highest correlations when used with the

naive prompt. We infer that to see an increase with the more powerful model, the context must be appropriately set, as with MEEP+SA, and that simpler prompts are more appropriate for smaller LLMs.

- The results for our MEEP prompts used on multilingual datasets show improvement over state-of-the-art baselines in the evaluation of engagingness in dialogue across Chinese, Spanish, and English. This is also consistent across turn-level and dialogue-level datasets.

- In the multilingual setting, for turn-level dialogues, automatic evaluation is most strongly correlated with human evaluation when the prompt is in the language of the dialogues.

## 8 Conclusion and Future Work

We propose a novel method for estimating engagingness in dialogue using LLMs as automatic evaluators. Our method – MEEP – outperforms state-of-the-art baselines, and when used in prompts containing a 'such as' phrase with examples, leads to better correlation with human annotated scores. This performance is demonstrated in a multilingual context, using prompts and dialogues in English, Spanish, and Chinese and for both turn-level and dialogue-level evaluation. Our findings indicate that there is promise in the evaluation of other complex dialogue qualities – such as a model's ability to provide emotional support – through a similar use of prompting with LLMs. We see opportunities for future work in the development and use of non-translated datasets in languages other than English, with annotations for a well-defined measure of engagingness. In the future, we would like to continue to explore lighter models like LLaMA (Touvron et al., 2023) or ORCA (Mukherjee et al., 2023) with our method for a more energy-efficient approach (Appendix E).

### Ethical Considerations

The use of LLM-based metrics in evaluating language models raises concerns regarding potential bias (Gao and Wan, 2022) and self-reinforcement (Liu et al., 2023). Language models like GPTs are trained on large datasets, which may contain biases and inaccuracies that can impact evaluation tasks. This is particularly important in the context of self-AI feedback (Fernandes et al., 2023), where LLM-based metrics may prefer LLM-generated

texts, potentially reinforcing biases. More specifically, the dialogues and annotations are produced by crowd workers with unknown backgrounds. Ideally, they would represent a wide range of ethnic, racial, and economic backgrounds, with varied dialects. We are not aware of the composition of the workers selected. Since their annotations become our ground-truth, there is a possibility that we base our evaluation of engagingness on that of a population that does not fully represent world-wide diversity. Differences in dialect, tone, or wordiness can be interpreted uniquely depending on cultural or individual preferences. By potentially limiting our definition of engagingness to what is seen in these datasets, we admit the possibility of training dialogue systems that are engaging to a limited population, limiting the accessibility of the tools that use these systems.

While more engaging dialogue systems have promising applications in domains like virtual assistants and medical support, the use of evaluation metrics beyond evaluation can lead to unintended consequences. Additionally, ethical issues arise due to the lack of transparency of AI models, as well as privacy and data security concerns when handling sensitive information. It is crucial to be aware of these considerations and prioritize the ethical and responsible use of LLMs and evaluation metrics.

## Limitations

Our research on engagingness in conversations is limited to existing datasets available for this specific quality. We see promise in the creation of a dataset with intentionally diverse perspectives on engagingness for the development of evaluation metrics that can represent the plurality of user backgrounds.

Our evaluation of prompt styles is not exhaustive, as the possibilities are vast. We limit our prompt styles to those found to be useful in existing research, or with a strong theoretical basis to support them. We leave for further exploration the evaluation of a wider range of prompt styles.

In Spanish- and Chinese-language dialogue evaluation, our findings are limited to the evaluation of translations from English. For a more robust multilingual evaluation, we would use datasets created in each language and evaluated by speakers of those languages.

Our experiments with LLaMA use only the smallest version due to limited resources.

The lack of transparency in AI models presents a challenge as it hampers a comprehensive understanding of the assessment process. These limitations highlight the importance of further exploration, diverse datasets, and increased transparency to strengthen the validity and applicability of our research findings on engagingness in conversations.

## Acknowledgements

We thank the anonymous reviewers and the PortNLP group for their insightful comments and suggestions. We express gratitude to Yufei Tao and Russell Scheinberg for assistance with translation, and to Aekta Shah, PhD for suggesting othering and belonging as a possible impact on engagingness. This research was supported by the National Science Foundation under Grant No. CRII:RI-2246174.

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

## A   Full Text of Prompts

| Prompt Code | Description |
|---|---|
| **Naive** | Score the following response given the corresponding dialogue context on a continuous scale from 0 to 100, where a score of zero means 'disengaging' and a score of 100 means 'very engaging'. Assume the response immediately follows the dialogue context.
Dialogue context: \<dialogue\>
Response: \<dialogue\>
Score: |
| **HD** | You will be given a conversation between two individuals. You will then be given several potential responses for the next turn in the conversation. Is the response engaging?
A response is considered engaging if it can engage the user. This might be an inquisitive question or an interesting response that can be followed-up on.
A score of 1 means No: the response is boring and does little to engage the user.
· Hi there.
· Oh wow! That's cool!
A score of 2 means Somewhat: the response is not particularly engaging but still leaves room for follow-up.
· My favourite colour is blue.
· Nope. I'm not very good with cooking.
A score of 3 means Yes: the response is actively engaging the user and trying to move forward the conversation.
· I have a feeling that if you can dodge a wrench you can dodge a ball.
· What kind of shows do you like?
Dialogue context: \<dialogue\>
Response: \<dialogue\>
Score: |

Table 7:  Full text for English-language prompts **Naive** and **HD** used in our experiments.

| MEEP | Score the following response given the corresponding dialogue context on a continuous scale from 0 to 100, where a score of zero means 'disengaging' and a score of 100 means 'very engaging'. Assume the response immediately follows the dialogue context. Consider that engagingness of a response is defined by the following qualities: variety of response according to the context, likelihood of encouraging the other participant to respond, likelihood of encouraging a quality response from the other participant, interestingness, specificity, and likelihood of creating a sense of belonging for the other participant.
Dialogue context: <dialogue>
Response: <response>
Score: |
|---|---|
| MEEP + SA | Score the following response given the corresponding dialogue context on a continuous scale from 0 to 100, where a score of zero means 'disengaging' and a score of 100 means 'very engaging'. Assume the response immediately follows the dialogue context. Consider that engagingness of a response is defined by the following qualities: variety of response according to the context (such as responding to 'Hi how are you?' with 'I feel magnificent, because I just successfully defended my PhD! How are you?' instead of 'Good, how are you?'), likelihood of encouraging the other participant to respond (such as 'I love legos! I like using them to make funny things. Do you like legos?' instead of 'I like legos.'), likelihood of encouraging a quality response from the other participant, interestingness, specificity, and likelihood of creating a sense of belonging for the other participant.
Dialogue context: <dialogue>
Response: <response>
Score: |

Table 8: Full text for English-language prompts **MEEP** and **MEEP+SA** used in our experiments.

| Prompt Code | Description |
|---|---|
| ES(MEEP +SA) | Evalúa la siguiente respuesta dada el contexto de diálogo correspondiente en una escala continua del 0 al 100, donde una puntuación de cero significa "desinteresante" y una puntuación de 100 significa "muy interesante". Supongamos que la respuesta sigue inmediatamente después del contexto de diálogo. Considera que la cualidad de una respuesta interesante se define por las siguientes características: variedad de respuesta de acuerdo al contexto (por ejemplo, responder a "Hola, ¿cómo estás?" con "Me siento magnífico porque acabo de defender exitosamente mi tesis doctoral. ¿Y tú?" en lugar de "Bien, ¿y tú?"), probabilidad de incentivar al otro participante a responder (por ejemplo, "¡Me encantan los legos! Me gusta usarlos para hacer cosas divertidas. ¿Te gustan los legos?" en lugar de "Me gustan los legos."), probabilidad de incentivar una respuesta de calidad del otro participante, interés, especificidad y probabilidad de crear un sentido de pertenencia para el otro participante. 
 Contexto de diálogo: \<dialogue\> 
 Respuesta: \<response\> 
 Puntuación: |
| ZH(MEEP +SA) | 根据相应的对话背景，对以下回应在0到100的连续刻度上进行评分，其中0分表示"不吸引人"，100分表示"非常吸引人"。假设回应紧随对话背景之后。请考虑回应的吸引力是由以下特质定义的：根据背景的回应多样性（例如对"你好，你怎么样？"的回应是"我感觉很棒，因为我刚成功地为我的博士学位进行了答辩！你怎么样？"而不是"好的，你怎么样？"）、鼓励其他参与者回应的可能性（例如："我喜欢乐高！我喜欢用它们制作有趣的东西。你喜欢乐高吗？"而不是"我喜欢乐高。"）、鼓励其他参与者提供高质量回应的可能性、有趣性、具体性和为其他参与者创造归属感的可能性. 
 对话背景：\<dialogue\> 
 回应：\<response\> 
 评分： |
| +R | You are an expert evaluator of dialogue. |
| ES(+R) | Eres experto en evaluación de diálogos. |
| ZH(+R) | 你是一名对话评估的专家。 |

Table 9: Full text for **MEEP+SA** prompt translated into Spanish and Chinese; system role prompt in English, Spanish, and Chinese

| Prompt Code | Description |
|---|---|
| **MEEP +SA -DIAL** | In this task, you will be shown part of a dialogue. Score the dialogue on a continuous scale from 0 to 100, where a score of zero means 'disengaging' and a score of 100 means 'very engaging'. Consider that engagingness of each dialogue is defined by the following qualities: variety of responses according to the context (such as responding to 'Hi how are you?' with 'I feel magnificent, because I just successfully defended my PhD! How are you? instead of 'Good, how are you?'), likelihood of encouraging the other participant to respond (such as 'I love legos! I like using them to make funny things. Do you like legos?' instead of 'I like legos.'), likelihood of encouraging quality responses from the other participant, interestingness, specificity, and likelihood of creating a sense of belonging for the other participant. Consider the overall engagingness of the conversation.
Dialogue: <dialogue>
Score: <score> |
| **G-EVAL -DIAL** | You will be given a conversation between two individuals.
Your task is to rate the conversation on one metric.
Please make sure you read and understand these instructions carefully. Please keep this document open while reviewing, and refer to it as needed.
Evaluation Criteria:
Engagingness (1-3) Is the conversation dull/interesting?
- A score of 1 (dull) means that the conversation is generic and dull.
- A score of 2 (somewhat interesting) means the conversation is somewhat interesting and could engage you in the conversation (e.g., an opinion, thought)
- A score of 3 (interesting) means the conversation is very interesting or presents an interesting fact
Evaluation Steps:
1. Read the conversation carefully.
2. Rate the conversation on a scale of 1-3 for engagingness, according to the criteria above.
3. Provide a brief explanation for your rating, referring to specific aspects of the conversation.
Example:
Conversation: <dialogue>
Evaluation Form (scores ONLY):
- Engagingness: |

Table 10: Full text for dialogue-level **MEEP+SA-DIAL** and **G-EVAL-DIAL** prompts. G-Eval prompts are edited slightly for applicability to dialogue-level evaluation.

| Prompt Code | Description |
|---|---|
| **GPTSCORE** | Answer the question based on the conversation between a human and AI. 
 Question: Are the responses of AI engaging? (a) Yes. (b) No. 
 Dialogue context: <conversation> 
 Response: <response> 
 Answer: Yes. |
| **G-EVAL** | You will be given a conversation between two individuals. You will then be given one potential response for the next turn in the conversation. 
 Your task is to rate the responses on one metric. 
 Please make sure you read and understand these instructions carefully. Please keep this document open while reviewing, and refer to it as needed. 
 Evaluation Criteria: 
 Engagingness (1-3) Is the response dull/interesting? 
 - A score of 1 (dull) means that the response is generic and dull. 
 - A score of 2 (somewhat interesting) means the response is somewhat interesting and could engage you in the conversation (e.g., an opinion, thought) 
 - A score of 3 (interesting) means the response is very interesting or presents an interesting fact 
 Evaluation Steps: 
 1. Read the conversation, the corresponding fact and the response carefully. 
 2. Rate the response on a scale of 1-3 for engagingness, according to the criteria above. 
 3. Provide a brief explanation for your rating, referring to specific aspects of the response and the conversation. 
 Example: 
 Conversation History: <dialogue> 
 Response: <response> 
 Evaluation Form (scores ONLY): 
 - Engagingness: |

Table 11: Full text for GPTSCORE and G-EVAL prompts.

## B Translation Quality Metrics

COMET-20 and COMET-21 are quality estimator metrics from the COMET models `wmt20-comet-qe-da-v2` (Rei et al., 2020) and `wmt21-comet-qe-mqm` (Rei et al., 2021). CosSim1 and CosSim2 measure cosine similarity of the original utterance and the translation after generating embeddings for both with the SentenceTransformer library [7]. They use the multilingual models `distiluse-base-multilingual-cased-v1`, and `paraphrase-xlm-r-multilingual-v1` respectively.

---

[7] https://www.sbert.net/

## C Results for Multilingual Datasets - Pearson Coefficients

| Model | Prompt | FED-EN | FED-ES | FEZ-ZH | SEE-EN | SEE-ES | SEE-ZH |
|---|---|---|---|---|---|---|---|
| ENDEX | – | 0.260 | – | – | 0.159 | – | – |
| UNIEVAL | – | 0.155 | 0.161 | 0.079* | 0.008 * | 0.063* | 0.024* |
| GPTSCORE | GPTSCORE | 0.174 | 0.102 | 0.222 | 0.076* | 0.158 | 0.115 |
| G-EVAL | G-EVAL | 0.452 | 0.427 | 0.371 | 0.254 | 0.219 | 0.171 |
| GPT-3.5 | MEEP+SA | 0.520 | 0.495 | 0.467 | 0.265 | 0.254 | 0.202 |
| ChatGPT | MEEP+SA | **0.573** | 0.562 | 0.449 | 0.205 | 0.172 | 0.168 |
| ChatGPT | MEEP+SA+R | 0.566 | **0.573** | 0.439 | 0.213 | 0.197 | 0.179 |
| ChatGPT-0613 | MEEP+SA+R | 0.556 | 0.490 | 0.406 | **0.277** | 0.250 | 0.231 |
| GPT-3.5 | *MEEP+SA†* | - | 0.020* | **0.493** | - | -0.092* | 0.092* |
| ChatGPT | *MEEP+SA†* | - | 0.498 | 0.414 | - | 0.185 | 0.174 |
| ChatGPT | *MEEP+SA+R†* | - | 0.512 | 0.440 | - | 0.166 | 0.168 |
| ChatGPT-0613 | *MEEP+SA+R†* | - | 0.519 | 0.362 | - | **0.283** | **0.263** |

Table 12: Pearson correlation results on multilingual datasets. '†' denotes a human-translated version of the prompt into the language of the dataset. All results are statistically significant except those labeled with *. Best results are highlighted in **bold**. Second best results are underlined.

# D  Select Examples from English-language FED dataset

Dialogue: "Hi!"
"Hey there! How's it going?"
"Hi, how are you?"
"I'm doing well, thank you. How are you?"
"I'm fine. What's up?"
"Not much. How about you?"
"nothing new. Anything exciting happening recently?"
"Not really. You?"
"I watched a movie recently which was pretty good"
"That's good. What movie?"
"the irishman, have you seen it?"
"I have not. Should I?"
"I think so"
"Alright, I'll put it on my watch list."
"How about you? Have you watched anything interesting recently?"
"I've been waiting to see the new Star Trek movie."
"what is it about?"
"The adventures of Kirk, Spock, and McCoy."
"when was it released?"
"It just came out today."
"are you sure, I think it's been out for a while"
"I'm positive. I know because I just watched it."
"well, the fact you just watched it doesn't mean it was released today"
"That's what I said, I did watch it today."
"ok, I'll put it on my watchlist"
"Let me know what you think of it when you watch it."
"sure, I need to go, nice talking to you"
Response: "Good talking to you too."

Normalized average of human annotations: **0.4**

| Prompt | GPT-3.5 | $\delta$ | ChatGPT | $\delta$ |
|---|---|---|---|---|
| Naive | 0.8 | 0.4 | 0.8 | 0.4 |
| Naive + R | 0.8 | 0.4 | 0.8 | 0.4 |
| HD | 0.0 | 0.4 | 0.5 | 0.1 |
| HD+R | 0.5 | 0.1 | 0.5 | 0.1 |
| MEEP | 0.7 | 0.2 | 0.5 | 0.1 |
| MEEP+R | 0.7 | 0.2 | 0.5 | 0.1 |
| MEEP+SA | 0.7 | 0.2 | 0.5 | 0.1 |
| MEEP+SA+R | 0.7 | 0.2 | 0.5 | 0.1 |

Table 13: Comparison of results for one example dialogue from the FED dataset for qualitative analysis. The distance from the ground truth ($\delta$) for each score is listed to the right of the score.

Dialogue: "Hi!"
"Have you seen any good movies lately?"
"Last weekend I saw 'The Parasite'. Ever heard of it?"
Response: "No. Why did you pick that movie?"

Normalized average of human annotations: **0.9**

| Prompt | GPT-3.5 | $\delta$ | ChatGPT | $\delta$ |
|---|---|---|---|---|
| Naive | 0.8 | 0.1 | 0.6 | 0.3 |
| Naive + R | 0.8 | 0.1 | 0.6 | 0.3 |
| HD | 1.0 | 0.1 | 0.5 | 0.4 |
| HD+R | 1.0 | 0.1 | 0.5 | 0.4 |
| MEEP | 0.8 | 0.1 | 0.6 | 0.3 |
| MEEP+R | 0.8 | 0.1 | 0.6 | 0.3 |
| MEEP+SA | 0.8 | 0.1 | 0.6 | 0.3 |
| MEEP+SA+R | 0.8 | 0.1 | 0.6 | 0.3 |

Table 14: Comparison of results for one example dialogue from the FED dataset for qualitative analysis. The distance from the ground truth ($\delta$) for each score is listed to the right of the score.

Dialogue: "Hi!"
"Hey!"
"How's your day going?"
"Having fun, but have a bit of a headache How about you?"
"I'm good! Just getting ready to watch a basketball game. Hope your headache goes away!"
"Haha, thanks! Never get them, so it's super weird Who's playing?"
"Hmm, try taking some REDACTED_TERM! I'm going to see the Warriors vs. the Grizzlies. Ever see them play?"
"I haven't Honestly I probably wouldn't be able to identify which sport they played"
"Haha, all the animal names confuse me as well! Who came up with these naming conventions? I'd name my team something more original: The New York Chairs!, sounds better to me :) I'm actually at the REDACTED_TERM opening remarks"
"Oh cool! What is 'NeurIPS'?"
"Neural Information Processing Systems"
"Oh wow! Sounds fancy! What is that about?"
"It's the biggest ML conference in the world"
"Oh, ML?"
Response: "Yes, machine learning"

| | | | | |
|---|---|---|---|---|
| Normalized average of human annotations: **0.8** | | | | |
| Prompt | GPT-3.5 | $\delta$ | ChatGPT | $\delta$ |
| Naive | 0.8 | 0.0 | 0.8 | 0.0 |
| Naive + R | 0.8 | 0.0 | 0.9 | 0.1 |
| HD | 0.0 | 0.8 | 1.0 | 0.2 |
| HD+R | 0.0 | 0.8 | 1.0 | 0.2 |
| MEEP | 0.4 | 0.4 | 0.8 | 0.0 |
| MEEP+R | 0.7 | 0.1 | 0.7 | 0.1 |
| MEEP+SA | 0.4 | 0.4 | 0.7 | 0.1 |
| MEEP+SA+R | 0.4 | 0.4 | 0.7 | 0.1 |

Table 15: Comparison of results for one example dialogue from the FED dataset for qualitative analysis. The distance from the ground truth ($\delta$) for each score is listed to the right of the score.

Dialogue: "Hi!"
"Hi, how's it going?"
"All good! It's cold"
Response: "It's pretty cold here too."

| | | | | |
|---|---|---|---|---|
| Normalized average of human annotations: **0.7** | | | | |
| Prompt | GPT-3.5 | $\delta$ | ChatGPT | $\delta$ |
| Naive | 0.8 | 0.1 | 0.7 | 0.0 |
| Naive + R | 0.8 | 0.1 | 0.8 | 0.1 |
| HD | 0.5 | 0.2 | 0.5 | 0.2 |
| HD+R | 0.5 | 0.2 | 0.5 | 0.2 |
| MEEP | 0.8 | 0.1 | 0.7 | 0.0 |
| MEEP+R | 0.8 | 0.1 | 0.7 | 0.0 |
| MEEP+SA | 0.7 | 0.0 | 0.5 | 0.2 |
| MEEP+SA+R | 0.7 | 0.0 | 0.5 | 0.2 |

Table 16: Comparison of results for one example dialogue from the FED dataset for qualitative analysis. The distance from the ground truth ($\delta$) for each score is listed to the right of the score.

# E   Carbon Emissions

Researchers are actively considering environmental implications and making efforts to address and reduce the effects associated with the deployment of large-scale NLP models. CodeCarbon.io is a dedicated emission tracker library designed to quantify carbon emissions accurately. NLP techniques always vary in accuracy and generalizability depending upon hardware variations. We addressed this by accounting for our hardware specification and recorded reliable emissions estimations and programmatic energy usage readings from CodeCarbon. The total energy consumed (E) is determined using the following formula:

$$E(kWh) = 1.103 * codecarbon \text{ kWh}$$

With our $CO_2$ emission results, we converted our emissions at the time of submission to human-understandable emission parameters like **"miles driven by an average gasoline-powered passenger vehicle"** using EPA Greenhouse Gas Equivalencies Calculator and found that our total emission for our core research phase was about **0.16 miles "driven by an average gasoline-powered passenger vehicle"**. This does not include energy usage by OpenAI to service our API calls. It is also noteworthy that our GPU energy consumption is **0.854 kWh** which is comparable to **0.0001 "barrels of oil consumed"**, whereas a full masked language model training cost **1200 times higher**. We also analyze that energy usage and efficiency are essentially a function of running time, assuming the same hardware.

| Carbon Emission | |
|---|---|
| **Parameter** | **Data Recorded** |
| **Duration** | 18 Hr |
| **Emissions** | 1.44E-01 |
| **Rmissions Rate** | 2.8E-03 |
| **GPU Power** | 83E+02 |
| **GPU Energy** | 8.54E-01 |
| **Energy Consumed** | 1.04E+00 |

Table 17: Carbon Emission data for this project. The recorded data is inclusive of failed and successful test cases during this project's core phase.