# OpenReview forum: "MEEP: Is this Engaging? Prompting Large Language Models for Dialogue Evaluation in Multilingual Settings"
_EMNLP/2023/Conference — EMNLP 2023 Findings_

### Official Review · Reviewer_F5G1 · 2023-08-04

**Soundness:** 3

**Excitement:**

2: Mediocre: This paper makes marginal contributions (vs non-contemporaneous work), so I would rather not see it in the conference.

**Paper Topic And Main Contributions:**

This paper proposes large language models with prompt to evaluate the engagingness of dialogue in multilingual setting.


**Reasons To Accept:**

This paper proposes large language models with prompt to evaluate the engagingness of dialogue in multilingual setting.  Some discussions may have insight for future research.

**Reasons To Reject:**

1. This work is limited to turn-level evaluation of engagingness, and there is no more valuable dialogue-level evaluation.
2. This paper is about multilingual evaluation, but in fact, only English translation to Spanish or Chinese evaluation is conducted, so the results and conclusions are not necessarily reliable.  In addition, the styles of prompt are relatively single, and in fact, it can be attempted to evaluate more styles.
3. Table 4 and Table 9 lack some results, but the authors did not provide convincing reasons.
4. The novelty of the method for measuring engagingness in dialogue is relatively limited, and is basically prompt engineering.

**Reproducibility:**

3: Could reproduce the results with some difficulty. The settings of parameters are underspecified or subjectively determined; the training/evaluation data are not widely available.

**Reviewer Confidence:**

4: Quite sure. I tried to check the important points carefully. It's unlikely, though conceivable, that I missed something that should affect my ratings.

---

> ### Author Rebuttal · Authors · 2023-08-29
>
> ### TLDR:
>
> Included more experiments in dialogue-level, reported all prompt styles used, explained novelty of our work and reported new results.
>
> &nbsp;
> ### Our Responses:
>
> Thank you for your insightful comments.
>
> + “This work is limited to turn-level evaluation of engagingness, and there is no more valuable dialogue-level evaluation.”:
>     + **Our response :**
> There is limited availability of datasets with annotations for engagingness at the dialogue-level for languages other than English. Nonetheless, the flexibility of our proposed method allows easy generalization to dialogue-level datasets. We have run new experiments and report the results below. The datasets have human annotations of engagingness for Chinese-language dialogues  (DSTC11, 2023). Each dataset has also been translated into English by the dataset authors. KDCONV-EN and KDCONV-ZH have 22 human-annotated dialogues. LCCC-EN and LCCC-ZH each have 295 human-annotated dialogues. The datasets are publicly available.  Our proposed method outperforms the baseline model (G-Eval) on 3 out of 4 datasets. We can easily integrate these results in the revised version.
>
> | Model | Prompt | KDCONV-EN | KDCONV-ZH | LCCC-EN | LCCC-ZH|
> | ------ | ------ |------ | ------ |------ | ------ |
> | G-Eval |--|**0.327***|0.189  |0.149| 0.248|
> | GPT3.5 |MEEP+SA|0.286*|0.223*|**0.282**|**0.287**
> | ChatGPT |MEEP+SA |0.178*|0.392*|0.073*|0.191
> | ChatGPT |MEEP+SA+R|0.185*|**0.428**|-0.004*|0.193
> | ChatGPT-0613 |MEEP+SA+R|0.186*|0.362*|0.032*|0.117
>
> Caption: Correlation results using Spearman coefficient for dialogue-level experiments. * denotes results that are not statistically significant.
>
> &nbsp;
> + “This paper is about multilingual evaluation, but in fact, only English translation to Spanish or Chinese evaluation is conducted, so the results and conclusions are not necessarily reliable”:
>
>     +   **Our response :** To our knowledge, datasets with annotations for engagingness are only available in these languages with translations. Use of translated datasets is standard in this setting and has been done in contemporary studies (Mendonca et al., 2023; Platek et al., 2023; DSTC11, 2023).
>
>
> &nbsp;
> + “The styles of prompt are relatively single, and in fact, it can be attempted to evaluate more styles.”:
>
>     + **Our response :**
>         We have evaluated many styles and included the 12 that are most effective. The evaluated styles presented in the paper include
>         + a Naive prompt,
>         + a Naive prompt with system role assignment,
>         + a Human-Directed prompt,
>         + a Human-Directed prompt with system role assignment,
>         + a prompt including our MEEP dimensions,
>         + a prompt including our MEEP dimensions and system role assignment,
>         + a prompt including our MEEP dimensions and some examples using ‘such as’,
>         + a prompt including our MEEP dimensions and some examples using ‘such as’, all translated into Chinese for use on Chinese dialogues,
>         + a prompt including our MEEP dimensions and some examples using ‘such as’, all translated into Spanish for use on Spanish dialogues,
>         + a prompt including our MEEP dimensions and some examples using ‘such as’ with system role assignment,
>         + a prompt including our MEEP dimensions and some examples using ‘such as’ with system role assignment translated into Chinese for use on Chinese dialogues, and
>         + a prompt including our MEEP dimensions and some examples using ‘such as’ with system role assignment translated into Spanish for use on Spanish dialogues.
>
>     The selection of this set of prompts was careful to include a range of styles commonly used from the simple and straight-forward Naive prompt, to the often-used adaptation of Human-Directed prompts (Gao et al. 2023; Liu et al. 2023), system role assignment, and the previously successful use of ‘such as’ as seen in other work (Yuan et al. 2021 (BARTScore)). It is important to evaluate these untested methods to provide evidence to confirm or deny the validity of their use. We contribute MEEP – our careful multidimensional definition of engagingness – as a successful new style for evaluation of engagingness in dialogue. We additionally evaluate the effectiveness of prompting in the language of the dialogues and find that it improves the correlation of automatic evaluation with human judgment.
>
>
>
> &nbsp;
> + “Table 4 and Table 9 lack some results, but the authors did not provide convincing reasons.”
>      + **Our response :**
> Thank you for this comment. The reasons are:
>     	+ A) Rows 1-2, Column **Prompt**:
>     	There are no input prompts for EnDex and UniEval because they either have their own prompts or use a different method for evaluation.
> 	    + B) Row 1, Columns **FED-ES**, **FED-ZH**, **SEE-ES**, **SEE-ZH**:
> 	    EnDex isn’t able to process dialogues in other languages, as noted in Section 5.3.
>     	+ C) Rows 9-12, Columns **FED-EN**, **SEE-EN**:
>     	The prompts in these rows are translated into the language of the dialogue/response pairs (as noted in the table caption). For example, a Chinese-language prompt is used in the evaluation of Chinese-language dialogue/response pairs. English dialogue/response pairs don’t have corresponding translated prompts because the prompts are already in English. Results for this combination are shown in rows 5-8, columns FED-EN, and SEE-EN. Listing them in the lower section would be redundant.
>
> &nbsp;
> + “The novelty of the method for measuring engagingness in dialogue is relatively limited, and is basically prompt engineering.”
>      + **Our response :**
> Our research contributes novel methods of evaluation (an open problem in NLG) in a challenging problem (engagingness in dialogue) while also evaluating the emergent abilities of LLMs (Wei et al., 2022). Given a well-crafted prompt, we investigate whether these LLMs can be used to generate automatic evaluations of dialogue response engagingness.
> We address a unique challenge in the evaluation of engagingness – a particularly complex quality of dialogue – and present a solution that draws from interdisciplinary perspectives. Our carefully developed dimensions of engagingness produce results that surpass state of the art metrics as well as ‘basic/simpler’ prompts. Our research further develops limited prior work on comparing prompt-based methods of evaluation to other baselines.
>
> &nbsp;
> ___
> DSTC11. Dstc11: Dialogue system technology challenge 11. https://chateval.org/dstc11. [Accessed 06-Jun-2023].
>
> Mingqi Gao, Jie Ruan, Renliang Sun, Xunjian Yin, Shiping Yang, and Xiaojun Wan. 2023. Human-like summarization evaluation with chatgpt.
>
> Yang Liu, Dan Iter, Yichong Xu, Shuohang Wang Ruochen Xu, and Chenguang Zhu. 2023. G-eval: NLG evaluation using GPT-4 with better human alignment.
>
> John Mendonça, Patrícia Pereira, Helena Moniz, Joao Paulo Carvalho, Alon Lavie,  Isabel M. Trancoso. Simple LLM Prompting is State-of-the-Art for Robust and Multilingual Dialogue Evaluation (Track 4). Dialogue System Technology Challenge 11. 2023 (accepted).
>
> Ondřej Plátek, Ondrej Dusek, Patricia Schmidtova,  Vojtech Hudecek, Matheusz Lango. Three Ways of Using Large Language Models to Evaluate Chat. Dialogue System Technology Challege 11. 2023 (accepted).
>
> Jason Wei, Yi Tay, Rishi Bommasani, Colin Raffel, Barret Zoph, Sebastian Borgeaud, Dani Yogatama, Maarten Bosma, Denny Zhou, Donald Metzler, Ed H. Chi, Tatsunori Hashimoto, Oriol Vinyals, Percy Liang, Jeff Dean, William Fedus. Emergent Abilities of Large Language Models. Transactions on Machine Learning Research. 2022
>
> Weizhe Yuan, Graham Neubig, and Pengfei Liu. 2021. Bartscore: Evaluating generated text as text generation. In Advances in Neural Information Processing Systems, volume 34, pages 27263–27277. Curran Associates, Inc.

---

### Official Review · Reviewer_amui · 2023-08-04

**Soundness:** 3

**Excitement:**

2: Mediocre: This paper makes marginal contributions (vs non-contemporaneous work), so I would rather not see it in the conference.

**Paper Topic And Main Contributions:**

This paper offers a prompt framework for automatic dialogue engagingness evaluation. By identifying six dialogue engagement dimensions in prompt designing, the proposed method makes use of LLMs' multilingual dialogue understanding ability, and obtains higher correlation coefficients than previous approaches.

**Reasons To Accept:**

This paper proposes a thorough definition of engagingness with six dimensions based on previous studies, which gives the evaluation approach certain linguistic interpretability.

The ablation study disentangles the effects of different parts in prompt design (MEEP, HD, SR). The overall prompt style is clean.

The SCC and PCC results are impressive.This paper proposes a thorough definition of engagingness with six dimensions based on previous studies, which gives the evaluation approach certain linguistic interpretability.

The ablation study disentangles the effects of different parts in prompt design (MEEP, HD, SR). The overall prompt style is clean.

The SCC and PCC results are impressive.

**Reasons To Reject:**

There might be not enough theoretical discussion and in-depth analyses which help readers understand the prompt design. More motivations and insights are needed.

The engineering part might still need refinement.

* Considering that this work is all about evaluation, there might be a lack of experiments currently. It might be beneficial to conduct more evaluation experiments categorized by language types and dialog content.
* Although the style design is clean, the prompts are not well-organized (Table 6, 7). All sentences squeeze together.
* The Chinese translation of the proposed prompt (Table 7) is bad.

**Reproducibility:**

4: Could mostly reproduce the results, but there may be some variation because of sample variance or minor variations in their interpretation of the protocol or method.

**Reviewer Confidence:**

3: Pretty sure, but there's a chance I missed something. Although I have a good feel for this area in general, I did not carefully check the paper's details, e.g., the math, experimental design, or novelty.

---

> ### Author Rebuttal · Authors · 2023-08-29
>
> ### TLDR:
>
> Addressed more theoretical aspects of MEEP, explained results, improved translational quality and re-ran the experiment, also included dialogue level results.
>
> &nbsp;
> ### Our Responses:
>
> Thank you for your helpful feedback.
>
> + “There might be not enough theoretical discussion and in-depth analyses which help readers understand the prompt design. More motivations and insights are needed.”
>      +  **Our response :**
>     Thank you for this comment. We have cited our sources for the inspiration for each prompt style in section 4. The black box nature of these LLMs limits us in our ability to perform an in-depth analysis of why each prompt style works. Prompt engineering remains an active area of research, and we hope that our findings will pave the way for more robust, reliable, and flexible methods of evaluation for conversations not only in English but also other languages.
>
> &nbsp;
> + “Considering that this work is all about evaluation, there might be a lack of experiments currently. It might be beneficial to conduct more evaluation experiments categorized by language types and dialog content”:
>     + **Our response :**
>         While more evaluation is certainly always beneficial, what is currently possible is somewhat limited due to the number of suitable datasets that are available with annotations of engagingness in dialogue which are necessary for this study. Our research focuses on automatic evaluation of engagingness in dialogue which has received little attention. Our extensive evaluation considers 3 recent LLMs, 4 state-of-the-art baseline models, 12 distinct prompts, 6 benchmark datasets, and 3 languages.
>
> &nbsp;
> + “Although the style design is clean, the prompts are not well-organized (Table 6, 7). All sentences squeeze together. ”:
>     + **Our response :**
>     This can be easily fixed for the camera-ready version. Regarding the sentences squeezing together, the prompts are written in the paper exactly as they are input into the models. This is in paragraph form in some cases. We welcome more discussion about this.
>
> &nbsp;
> + “The Chinese translation of the proposed prompt (Table 7) is bad.”:
>     + **Our response :**
>
>     Thank you for this note. We have obtained a new translation from another expert and rerun relevant models to obtain updated results using the new translated prompt. In general, the results are slightly better using the new translation.
>     + The new translation is:
>
>          根据相应的对话背景，对以下回应在0到100的连续刻度上进行评分，其中0分表示“不吸引人”，100分表示“非常吸引人”。假设回应紧随对话背景之后。请考虑回应的吸引力是由以下特质定义的：根据背景的回应多样性（例如对“你好，你怎么样？”的回应是“我感觉很棒，因为我刚成功地为我的博士学位进行了答辩！你怎么样？”而不是“好的，你怎么样？”）、鼓励其他参与者回应的可能性（例如：“我喜欢乐高！我喜欢用它们制作有趣的东西。你喜欢乐高吗？”而不是“我喜欢乐高。”）、鼓励其他参与者提供高质量回应的可能性、有趣性、具体性和为其他参与者创造归属感的可能性.
>         对话背景：<dialogue>
>         回应：<response>
>         评分：
>
>     + The updated results are in the last four rows of Table 4, in the columns FED-ZH, and SEE-ZH:
>         + We see that the results for FED-ZH are 0.072 higher and results for SEE-ZH are 0.020 higher on average (omitting comparing statistically insignificant results) than those obtained with the previous translated prompt. For both datasets we see a new highest correlation. This indicates that the translated prompts provide better correlation than English-only prompts.
>
>             | Model | Prompt | FED-EN | FED-ES | FED-ZH |SEE-EN | SEE-ES | SEE-ZH|
>             | ------ | ------ |------ | ------ |------ | ------ |------ | ------ |
>             | ENDEX | -- |   0.290  | -- |   -- | 0.164 |  -- | -- |
>             | UNIEVAL |-- | 0.190 | 0.258| 0.076* | 0.015* | 0.073* | 0.038*
>             | GPTSCORE| GPTSCORE|0.176 | 0.146 | 0.230 | 0.087* | 0.153 | 0.140
>             | G-EVAL |  G-EVAL |0.488 | 0.448 | 0.402| 0.194 | 0.131 | 0.062*
>             | ------  ------ |------ | ------ |------ | ------ |------ | ------ |
>             | GPT3.5 | MEEP+SA | 0.532 | 0.481 | 0.451 |**0.236** | 0.223 |0.189
>             | ChatGPT |MEEP+SA | 0.558 | 0.516| 0.431 | 0.169 | 0.138| 0.133
>             | ChatGPT | MEEP+SA+R  |**0.568**|**0.542**| 0.435| 0.160 | 0.150 |0.140
>             | ChatGPT-0613 | MEEP+SA+R | 0.550 |0.471| 0.400| 0.214| 0.200 | 0.175
>             | ------  ------ |------ | ------ |------ | ------ |------ | ------ |
>             | GPT3.5 |MEEP+SA †|--|0.438  |**0.520**| --| 0.128 | 0.085*
>             | ChatGPT | MEEP+SA †| --| 0.500 | 0.408| -- | 0.161 | 0.149
>             | ChatGPT |  MEEP+SA+R †| --|0.525 | 0.444 | -- |0.123 | 0.168
>             | ChatGPT-0613 |  MEEP+SA+R †| --|**0.542**| 0.374| -- |**0.273**|**0.227**
> Caption: Correlation results using Spearman coefficient for multilingual experiments. All results are statistically
> significant except those labeled with *. ‘†’ denotes a human-translated version of the prompt into the language of
> the dataset.

---

### Official Review · Reviewer_ppYc · 2023-08-12

**Soundness:** 3

**Excitement:**

3: Ambivalent: It has merits (e.g., it reports state-of-the-art results, the idea is nice), but there are key weaknesses (e.g., it describes incremental work), and it can significantly benefit from another round of revision. However, I won't object to accepting it if my co-reviewers champion it.

**Paper Topic And Main Contributions:**

Firstly, Authors describe engagingness as important factor as it highly correlates to sense of humane behaviour during conversation.
They propose these sub-dimensions of engagingness:
- Response Diversity
- Interactional Quality
- Sentiment
- Interestingness
- Othering
- Contextual Specificity

Authors believe, engagingness is really important metric for conversational AI. Their Auto LLM evaluation would be significant for the community.

MEEP(Metric for Engagingness Evaluation using Prompting) is prompting technique to measure engagingness in dialogues across multiple languages.

To test their effectiveness of technique, They run it in multilingual settings. For getting multilingual datasets, Authors use Microsoft Azure & Tencent services for translation on existing English datasets. Authors use COMET and Cosine similarity to measure translation quality.

**Reasons To Accept:**

# Strengths
- MEEP is essentially prompting technique, which achieves better Spearman and Pearson coefficient of about 0.05 than baselines. It can be interpreted as statistically significant correlation.
-  The method can also be used in conjunction to other prompting techniques like Such As and Role Assignment (SA, R) achieving better performance.
- Various baselines like GPTSCORE, G-EVAL  have used for comparison are meaningful and widely used by community.

**Reasons To Reject:**

# Weakness
- **Limited scope** of paper to specific 'Engagingness' evaluation.
- The evaluation is limited to **ChatGPT & GPT-3.5**, which may limit the generalisation of this method. Evaluation on open-source models like **Pythia, Llama** would beneficial to make the claims more general.
- **Sub-dimensions** of engagingness is subjective, would need more explanation.
- Translated datasets and code used for this evaluation technique has not been **made public**.

**Reproducibility:**

2: Would be hard pressed to reproduce the results. The contribution depends on data that are simply not available outside the author's institution or consortium; not enough details are provided.

**Reviewer Confidence:**

3: Pretty sure, but there's a chance I missed something. Although I have a good feel for this area in general, I did not carefully check the paper's details, e.g., the math, experimental design, or novelty.

---

> ### Author Rebuttal · Authors · 2023-08-29
>
> ### TLDR:
> Discussed each feedback from the reviewer with facts. Provided importance of the Engagingness in dialogue evaluation and its sub-dimensions, and reported performance of new results.
>
> &nbsp;
> ### Our Response:
>
> Thank you for your detailed feedback.
>
> + “Authors use Microsoft Azure & Tencent services for translation on existing English datasets”:
>     +  **Our response :**
>     We’d like to clarify that we did not generate these translations. These translations were provided by the original authors of the datasets. We used the existing publicly available datasets without any modifications.
> &nbsp;
>
> &nbsp;
> + “MEEP achieves a better Spearman and Pearson coefficient of about 0.05 than baselines”:
>     +  **Our response :**
>     The average improvement in the submitted version is 0.072 above baselines and our updated results yield improvement of 0.090 above baselines (referred in Table 4). We have calculated the difference between the best of our method (highlighted score) and the best of baseline results, and averaged these over the FED and SEE datasets.
>
> &nbsp;
> + “Limited scope to engagingness”:
>     +  **Our response :**
>     Recent research is focused on making “engaging conversation” with dialogue agents with deeper, open-ended interactions, and sustainable communication to give users a human-like experience. Engagingness is more complex than other dimensions such as fluency or understandability. Existing multi-dimensional metrics have had limited success measuring engagingness and a focused metric such as ours can be a way to develop this capability. Both UniEval and GPTScore, which we use as baseline metrics, are multidimensional metrics (including engagingness as one of the dimensions) that consistently showed lower performance   when compared to our proposed method.
>
> &nbsp;
> +  “The evaluation is limited to ChatGPT & GPT-3.5, which may limit the generalisation of this method. Evaluation on open-source models like Pythia, Llama would beneficial to make the claims more general”:
>       +  **Our response :**
>     We agree that evaluation of open-source models is beneficial, however, unfortunately, at the time of our research implementation and experiments, access to large-scale open-source models that we wanted to explore (such as Llama) was rather limited. We are currently evaluating newer open-source models and will be able to integrate these results into the revised version.
>
> &nbsp;
> + “Sub-dimensions of engagingness is subjective, would need more explanation.”:
>      +  **Our response :**
>  We can certainly clarify this in the paper. It is worth noting that definitions of engagingness are often subjective, a factor that's commonly overlooked in research. To address this, we performed statistical analysis, as illustrated in Figure 2, focusing on the most prevalent sub-dimensions of dialogue evaluation metrics alongside engagingness. Notably, we opted for "interestingness" as an additional sub-dimension due to its strong correlation with engagingness, a choice substantiated by inputs from linguistics, dictionary definitions, and relevant semantic sources cited in our paper. The difficulty of defining engagingness as a quality is in part why it is such a valuable metric, as other less nuanced qualities are easier to train for and evaluate and have accordingly made up a significant portion of prior progress.
>
> &nbsp;
> + “Translated datasets and code used for this evaluation technique has not been made public.”
>     +  **Our response :**   Both datasets (FED and SEE) and their translations are publicly available (DSTC11, 2023). Our code, including identification of the subset of examples used for testing from the Persona-See datasets, will be made available upon acceptance of our paper.
>
> &nbsp;
> ___
> DSTC11. Dstc11: Dialogue system technology challenge 11. https://chateval.org/dstc11. [Accessed 06-Jun-2023].

---

### Meta-Review · Area_Chair_DryX · 2023-10-06

**Recommendation:** 2

**Metareview:**

While the paper presents promising results in terms of evaluation of engagingness, addressing concerns related to theoretical depth, prompt refinement, diverse experiments with additional models, translation quality in Chinese,  and novelty would strengthen its overall contribution to the field.

---

### Decision · Program_Chairs · 2023-10-07

**Decision:**

Accept-Findings

**Comment:**

While the paper presents promising results in terms of evaluation of engagingness, addressing concerns related to theoretical depth, prompt refinement, diverse experiments with additional models, translation quality in Chinese,  and novelty would strengthen its overall contribution to the field.